# Multiple Sclerosis: Enzymatic Cross Site-Specific Recognition and Hydrolysis of H2A Histone by IgGs against H2A, H1, H2B, H3 Histones, Myelin Basic Protein, and DNA

**DOI:** 10.3390/biomedicines10081876

**Published:** 2022-08-03

**Authors:** Georgy A. Nevinsky, Valentina N. Buneva, Pavel S. Dmitrienok

**Affiliations:** 1Institute of Chemical Biology and Fundamental Medicine of the Siberian Division of Russian Academy of Sciences, Lavrentiev Ave. 8, 630090 Novosibirsk, Russia; buneva@niboch.nsc.ru; 2Pacific Institute of Bioorganic Chemistry, Far East Division, Russian Academy of Sciences, 690022 Vladivostok, Russia; paveldmt@piboc.dvo.ru

**Keywords:** human blood sera antibodies-abzymes, multiple sclerosis patients, catalytic abzymes, hydrolysis of H2A histone, IgGs against H2B, H1, H2A, H3, H4 histones, myelin basic protein, DNA, enzymatic cross recognition and hydrolysis

## Abstract

Histones have a paramount role in chromatin remodeling and gene transcription. Free histones are damage-associated molecules in the blood; administration of histones to animals drives systemic inflammatory and toxic effects. Myelin basic protein (MBP) is the most crucial component of the axon myelin-proteolipid sheath. Antibodies-abzymes with different enzymatic activities are very toxic and an essential feature of some autoimmune diseases. Electrophoretically homogeneous IgGs against H1, H2A, H2B, H3, H4, MBP, and DNA were derived from sera of multiple sclerosis (MS) patients by several affinity chromatographies. Using MALDI-TOFF mass spectrometry, it was shown that IgGs against H2A split H2A at 12 sites; the number of H2A hydrolysis sites by antibodies against other antigens is different: H1 (19), H2B (11), H3 (15), H4 (9), MBP (10), and DNA (23), and they only partly match. Thus, the complex formation polyreactivity and the enzymatic cross-activity of pernicious humans IgGs against five histones, MBP, and DNA have been shown for the first time. The data obtained indicate that the formation of such polyspecific-polyreactive abzymes, whose single active center can recognize and hydrolyze different substrates, can occur due to the formation of antibodies against hybrid antigenic determinants consisting of several histone protein sequences. IgGs with high affinity for DNA with DNase and protease activities may be antibodies against DNA-histone complex antigenic determinants, including protein and DNA sequences. Polyreactive IgGs-abzymes against MBP, five histones, and DNA with extended cytotoxicity can play a very negative role in the pathogenesis of multiple sclerosis and probably other different diseases.

## 1. Introduction

Antibodies (Abs) to chemically stable analogues of reaction transition states and natural autoantibodies with enzymatic activities are called abzymes (ABZs), and they are well described in the literature [1,2,3,4,5,6]. The spontaneous and stimulated by different antigens development of various autoimmune diseases (ADs) results in the synthesis of ABZs by B-cells against polysaccharides, lipids, peptides, proteins, DNAs, and RNAs and their complexes. In the blood sera of AD patients, many different ABZs were discovered directly against many specific antigens, mimicking chemical reaction transition states. Secondary anti-idiotypic auto-ABZs to active sites of several classical enzymes were also detected; their formation may account for using Jerne’s model of the anti-idiotypic network [7]. The appearance of Abs-ABZs in the blood sera of mammals is very reliable and the earliest indicator of the onset of the autoimmune (AI) diseases in humans and mammals [1,2,3,4,5,6]. To date, different abzymes (IgGs, IgA, and IgMs) splitting DNAs, RNAs [8,9,10,11,12], poly and oligosaccharides [13,14,15], various peptides, and proteins [16,17,18,19,20,21,22,23] have been found in the blood sera of patients with different ADs and several viral pathologies [1,2,3,4,5,6].

Some healthy humans sometimes produce antibody-abzymes having very low vasoactive intestinal peptide [16], thyroglobulin [18], and polysaccharide-hydrolyzing [13,14,15] activities. At the same time, the blood of healthy humans and patients suffering from specific pathologies demonstrating insignificant AI reactions usually lack ABZs [1,2,3,4,5,6]. Nonetheless, germline antibodies of some healthy humans and mammals could possess amyloid- and superantigen-directed enzymatic activities [24,25].

Myelin basic protein (MBP) is the major and vital protein of the myelin-proteolipid sheath of axons. It is believed that the development of multiple sclerosis is associated with the hydrolysis of proteins, including MBP of the myelin sheath of nerve tissues. The specific abzymes against MBP can hydrolyze the axon myelin sheath MBP, having an essential negative role in MS pathogenesis due to infringement of nerve impulse conduction [1,2,3,4,5,6,21,22].

Histones and their different modified forms hold a vital role in the functioning of chromatin. Free extracellular histones in the blood are detrimental proteins causing toxic impacts through inflammatory pathways and teamwork with Toll-like receptors [26]. ABZs hydrolyzing five histones (H1–H4) and MBP were detected in the blood of HIV-infected [21,22,27,28,29,30,31,32,33,34], SLE [35], and MS [36] patients, as well as mice with autoimmune experimental encephalomyelitis [37]. In AD patients, many antibodies to histones and DNA are directed against histone-DNA complexes emerging in the blood due to cell apoptosis [38]. Antibodies that hydrolyze DNA are cytotoxic. They penetrate through membranes of cells and nuclei, hydrolyze DNA of chromatin and induce cell apoptosis [2,3,4,5,6]. The catalytic cross-reactivity of ABZs against MBP, histones, and DNA is dangerous to humans and mammals because all five histones, due to apoptosis of cells, occur in human blood. Considering this, the analysis of possible enzymatic cross-reactivity of abs-abzymes against MBP, histones, and DNA is critical for analyzing the beginning and progress of multiple sclerosis.

For canonical enzymes, the situation is simple: one gene-one enzyme. Classical enzymes specific for various substrates usually catalyze only one chemical reaction [39,40,41]. From theoretical estimation, due to the V (D) J recombination, regions of unique DNA encoding variable domains of the Abs human immune system can create about 10^6^ B-lymphocytes against one antigen, producing antibodies to the same antigen with very different properties [42,43,44,45].

The unspecific complex formation of some antigens with antibodies against different foreign compounds is a widely distributed phenomenon [46,47,48,49]. As described to date, Abs-abzymes against different proteins usually also only split their particular proteins [1,2,3,4,5,6]. The first examples of Abs with enzymatic cross-reactivity were IgGs against MBP and five histones (H1–H4) from sera of HIV-infected patients [32,33,34]. However, an analysis of the possible polyreactivity of recognition-complexation and enzymatic cross-reactivity of antibodies against the five histones (H1–H4) themselves has not yet been carried out. We have suggested that if abzymes against histones H1, H2A, H2B, H3, and H4 possess enzymatic cross-reactivity with abzymes against MBP, they potentially can split not only their specific histone but also four other histones as well.

In addition, using monoclonal light chains (MLChs) showed that their DNA sequences are identical (88–100%) to the germlines of IgLV8 light chain genes of several described antibodies [50,51,52,53,54,55,56]. At the same time, some of them can hydrolyze MBP as serine-like and metalloproteases. In contrast, others combine structural elements of proteases and DNases in the same active center, exhibiting the activity of typical classical serine-like, metalloproteases, and DNases [50,51,52,53,54,55,56]. In the DNA sequences of such MLChs, sequences encoding amino acid residues responsible for recognizing protein and DNA substrates, chelation of metal ions as specific cofactors of classical metalloproteases, and DNases directly involved in catalysis were found [54,55,56]. The main antigens for which antibodies against DNA and histones are produced are their complexes, which appear in the blood due to cell apoptosis [38].

It is believed that in MS patients, the development of AI reactions can be stimulated by different viral or bacterial infections [1,2,3,4,5,6]. Some proteins of Epstein–Barr, influenza, herpes, polyoma, and other viruses were reported to be able to mimic human myelin proteins. First, the immune system produces antibodies against viral proteins, and then it fails and begins to synthesize auto-antibodies against human proteins. Since antibodies against DNA complexes with histones are capable of hydrolyzing MBP, which constantly appears in the blood of people, in addition to producing external activation of the development of multiple sclerosis by viral proteins, there may be internal factors for the development of this pathology. Abzyme-antibodies with multiple extended catalytic functions appear to be particularly toxic and harmful to humans.

Considering this, the question has arisen whether antibodies can be produced only against proteins and DNA sequences of their complexes. Alternatively, in the case of DNA-histones complexes, the question becomes whether antibodies can be produced not only against DNA, histones or MBP but also against antigenic determinants formed at the junction of protein sequences and fragments of DNAs bound with histones or MBP. As a result of the development of Abs against protein-nucleic antigenic determinants, it is possible to expect the formation of abzymes that hydrolyze both DNA and proteins. We tried to test this hypothesis in this work.

In this work, the study of the ability of the IgGs from MS patients against H1, H2A, H2B, and H3 histones, MBP, and DNA to hydrolyze H2A histone was performed for the first time. It was shown that abzymes against these proteins and DNA possess catalytic cross-reactivity. It has been demonstrated that ABZs against histones, MBP, and DNA can split H2A histone with different efficiency and in different sites, and they hydrolyze DNA.

## 2. Material and Methods

### 2.1. Chemicals, Donors, and Patients

All compounds used, including BrCN-activated Sepharose, five electrophoretically homogeneous human histones (H2A, H1, H2B, H3, and H4) and an equimolar mixture of five histones, were purchased from Sigma (St. Louis, MO, USA). Protein G-Sepharose and Superdex 200 HR 10/30 were purchased from GE Healthcare (GE Healthcare, New York, NY, USA). Human myelin basic protein was bought from the Center of Molecular Diagnostics and Therapy (DBRC, Moscow, Russia). Protein Sepharose columns containing immobilized histones and MBP were prepared according to the manufacturer’s protocol using BrCN-activated Sepharose, five individual histones or their mixture, and MBP.

Proof that IgG antibodies of multiple sclerosis patients hydrolyze five histones (H1–H4) and myelin basic protein was earlier attained using IgG preparations from the blood of 59 MS patients [21,22,31]. Patient medical characteristics are given in [31]. The diagnosis of multiple sclerosis was established by specialists of the Novosibirsk Medical University Multiple Sclerosis Center based on the McDonald classification [57]. The disease severity of all 59 MS patients was scored using Kurtzke’s Expanded Disability Status Scale (EDSS) [58]. The MS patients at entry had no infection symptoms. None of the multiple sclerosis patients had received any anti-disease therapies at the sample collection time six months before the study.

Patients with MS gave written consent to provide blood for scientific purposes. The local human ethics committee (Novosibirsk State Medical University, Novosibirsk, Russia; document number: 105-HIV; 07. 2010). This ethics committee has supported this investigation according to the guidelines of the Helsinki ethics committee.

In this study, we used an equimolar mixture of 15 of 59 IgGs described earlier [31] with relatively high activity in the hydrolysis of five histones, MBP, and DNA. The patient medical characteristics of 15 patients are shown in Appendix A.

### 2.2. Antibody Purification

Electrophoretically homogeneous preparations of antibodies were isolated from the blood plasma of 15 MS patients first by affinity chromatography of the plasma proteins on Protein G-Sepharose. These were then additionally purified by gel filtration on a column with Superdex 200 HR 10/30 as in [31,32,33,34]. Analysis of IgG for homogeneity was performed using SDS-PAGE: 4–17% gradient gels (0.1% SDS), and all proteins were visualized using silver staining as in [31,32,33,34]. They were passed through 0.1 μm Millex filters to protect IgGs from potential contaminations. After 5–7 days of storage at 4 °C for refolding, the preparations of IgGs were used in different assays.

Earlier, after SDS-PAGE of IgG antibodies, 3–4-mm cross-sections of the gel longitudinal slices were used to get eluates. Proteolytic activity was detected only in the eluates corresponding to the gel fragments containing IgGs. It was shown that the obtained IgGs of multiple sclerosis patients do not contaminate any canonical proteinases and DNases [31].

### 2.3. Affinity Chromatography of IgGs on MBP-Sepharose

To isolate IgGs against each of the individual histones and MBP, a mixture of 15 antibody preparations (IgG_mix_) exhibiting relatively high activity in the cleavage of five histones (H1–H4), MBP, and DNA was used [31]. Removal of IgGs against all five histones from electrophoretically homogeneous IgG_mix_ having no admixture of any canonical proteases and DNases was reached using MBP-Sepharose (immobilized human MBP), equilibrated in 20 mM Tris-HCl, pH 7.5 (buffer A). To isolate the IgG fraction against MBP, the column was first washed to zero optical density (A_280_) with buffer A. Then, nonspecifically adsorbed IgGs with low affinities for MBP were eluted from the MBP-Sepharose column using buffer A supplemented with 0.2 M NaCl. Finally, IgGs with high affinities for MBP were eluted specifically first by 3.0 M NaCl and then with 0.1 M glycine-HCl, pH 2.6. The fractions eluted with 3.0 M and glycine-HCl were subjected to additional purification from potentially possible admixtures of antibodies against 5 histones. These fractions were combined and passed twice through the column of histone5-Sepharose (immobilized mixture of five histones). The fraction of antibodies eluted from the histone5-Sepharose column upon the loading and washing with 5 mL of buffer A was designated and used as anti-MBP IgGs.

The fraction containing IgGs against five histones, which was eluted from MBP-Sepharose at loading and the column washing with 5 mL of buffer A, was subjected twice to repeated passing through this sorbent and then used to isolate IgGs against five individual histones. It was subjected to chromatography on histone5-Sepharose. Nonspecifically bound IgGs were eluted with 0.2 M NaCl, while anti-histone IgGs used 3.0 M NaCl and acidic buffer (pH 2.6). A mixture of these two fractions was further used to get IgGs against five individual histones. It was applied first on H2A-Sepharose containing immobilized H2A histone. The fractions eluted at loading in the case of each of the used sorbents were applied sequentially to the next one: H1-Sepharose, H2B-Sepharose, H3-Sepharose, and H4-Sepharose. All chromatographies were carried out similarly to that in the case of MBP-Sepharose. IgGs against H2A and H1–H4 histones were specifically eluted from each affinity sorbent with buffer, pH 2.6. These fractions were designated, respectively, as anti-H2A, anti-H1, anti-H2B, anti-H3, and anti-H4 IgGs.

### 2.4. Affinity Chromatography of IgGs on DNA-Cellulose

The final fractions were used to get anti-DNA IgGs of MS patients. They were applied on DNA-cellulose (5 mL, equilibrated with 20 mM Tris-HCl buffer, pH 7.5). After elution of Abs with a low affinity for DNA 0.2 M NaCl, anti-DNA IgGs were eluted first with 3.0 M NaCl and then with the acidic buffer. To get anti-DNA IgGs, the fractions with no affinity for MBP-Sepharose and histone5-Sepharose (eluted from these sorbents upon loading) were also passed twice through these two affinity sorbents. By combining these two factions, anti-DNA IgGs were obtained.

### 2.5. Proteolytic Activity Assay

The reaction mixtures (8–15 μL) contained 25 mM Tris-HCl (pH 7.5), 0.8–1.0 mg/mL H2A histone (or a mixture of histones) or 1.0 mg/mL myelin basic protein, and 0.01–0.15 mg/mL IgGs against one of five histones, H1–H4, or MBP. All mixtures were incubated for 1–24 h at 37 °C. The efficiency of histone H2A and MBP cleavage was analyzed by SDS-PAGE using 15% gels in the absence of dithiothreitol under nonreducing conditions as in [27,28,29,30,31,32,33,34,35]. Five histones, H1–H4, and MBP splitting products were revealed using gel staining with Coomassie Blue. The gels were scanned after staining and quantified as in [31] using Image Quant v5.2 software. The efficiency of H1-H2A-H4 or MBP hydrolysis was evaluated by reducing the protein content compared to the control experiment—incubation of histones or MBP in the absence of antibodies.

### 2.6. DNA Hydrolysis

The reaction mixtures (10–17 μL) contained: 4.0 mM MgCl_2_, 0.2 mM CaCl_2_, 20 mM Tris-HCl, pH 7.5, 10 μg/mL *pBluescript* supercoiled (sc)DNA plasmid, and 2–32 μg/mL IgGs, similar to [11,12]. After incubation for 1.0–4.5 h at 37 °C, 2.5 μL of loading buffer containing 1% SDS, 50 mM EDTA, pH 8.0, 30% glycerol, and 0.005% bromophenol blue was added to the reaction mixture. Electrophoresis was performed using 0.8% agarose gel until the bromophenol blue migrated in 2/3 of the gel. scDNA in the gel was stained with ethidium bromide (0.5 μg/mL, 1–2.5 min). The gels were imaged using a Gel Doc gel documentation system (Bio-Rad). The photographs of the gels were counted using the ImageQuant 5.2 program. The level of DNA-hydrolyzing activity of IgGs was determined by the degree of hydrolysis of the scDNA plasmid form to its relaxed form.

### 2.7. MALDI-TOF Analysis of Abs-Dependent H2A Histone Hydrolysis

H2A hydrolysis by antibodies against histones, MBP, and DNA was carried out using the Reflex III system (Bruker Company; Frankfurt, Germany) containing a 337-nm VSL-337 ND nitrogen laser with a 3 ns pulse duration. Mixtures (10–14 µL) containing 20 mM Tris-HCl (pH 7.5), 0.8 mg/mL H2A histone and 0.04–0.05 mg/mL IgG were incubated at 30 °C during 0–20 h. To 1.3 µL of the sinapinic acid matrix mixed with 1.3 µL of 0.2% trifluoroacetic acid, 1.3 µL of the solutions containing H2A histone before or after incubation with different IgGs against each of five histones, MBP, and DNA were added; 1.1–1.2 µL of these mixtures was air-dried after loading on the MALDI plates. All MALDI-TOFF mass spectra were calibrated using standard Bruker Daltonic protein mixtures II and I (Germany) in two calibration modes: internal or external. The analysis of peptide molecular masses corresponding to specific sites of H2A splitting by IgG against five histones, MBP, and DNA was performed using Protein Calculator v3.3 (Scripps Research Institute; La Jolla, CA, USA).

### 2.8. Analysis of Sequence Homology

The analysis of protein sequence homology between histones and MBP was performed using *the lalign* site (http://www.ch.embnet.org/software/LALIGN_form.html (accessed on 1 January 2008)).

### 2.9. Statistical Analysis

The results are presented using mean ± S.D. of 7–10 independent MALDI mass specters for each sample of H2A hydrolysis by IgGs against each of five histones, MBP, and DNA.

## 3. Results

### 3.1. Purification of Antibodies

To analyze the “average” site-specific cleavage of H2A by IgGs against histones, MBP, and DNA, we have obtained the mixture of equal amounts of fifteen IgG preparations (IgG_mix_), possessing relatively high activities in the splitting of H2A, MBP, and DNA. In this study, we used previously analyzed electrophoretically homogeneous IgG preparations from the sera of fifteen multiple sclerosis patients isolated by sequential chromatography of the blood plasma proteins first on Protein G-Sepharose using conditions allowing us to remove nonspecifically bound proteins [31,32,33,34]. Then, polyclonal IgG preparations were subjected to FPLC gel filtration under drastic conditions (pH 2.6), destroying immune complexes [31,32,33,34]. Any possible artefacts due to potentially possible traces of contaminating classical proteases and DNases were excluded earlier [31]. After SDS-PAGE of IgG_mix_ preparation, its proteolytic and DNase activities in histones, MBP, and DNA hydrolysis were found in only one IgG_mix_ protein band [31].

### 3.2. Isolation of IgGs against MBP

IgGs against MBP were isolated from polyclonal IgG_mix_ preparation by affinity chromatography on MBP-Sepharose. The IgG fraction nonspecifically bound and having low affinity to MBP-Sepharose was eluted with 0.2 M NaCl. Specific anti-MBP IgGs with high affinities for MBP were first eluted using 3.0 M NaCl and then with Tris-Gly buffer, pH 2.6. For additional purification of anti-MBP IgGs from potential impurities of IgGs against five histones, the anti-MBP IgG fraction was passed through histone5-Sepharose (immobilized 5 histones). The fraction obtained at loading onto histone5-Sepharose and washing with 5 mL of buffer A was further used as anti-MBP IgGs.

### 3.3. Isolation of IgGs against Individual Histones

The IgGs eluted from MBP-Sepharose on loading and washing with 5 mL of buffer A (containing Abs against five histones) were combined and passed twice through MBP-Sepharose and then used to obtain IgG fractions against five individual histones. For this, polyclonal IgGs containing five histones were first applied on histone5-Sepharose and eluted specifically with 3.0 M NaCl and acidic buffer. Then, the fraction obtained was sequentially applied to four sorbents with immobilized five individual histones: H2A, H1, H2B, H3, and H4. The fraction eluted upon loading without affinity for the previous sorbent was used for following subsequent chromatographies. Finally, we prepared five IgG preparations against five individual histones: anti-H2A, anti-H1, anti-H2B, anti-H3, and anti-H4 IgGs.

IgGs eluted from MBP-Sepharose and histone5-Sepharose during loading were additionally passed twice through these sorbents and then applied to DNA-cellulose. Antibodies with a high affinity for DNA-cellulose were eluted from this sorbent with 3.0 M NaCl and acidic buffer and used as anti-DNA IgGs.

### 3.4. SDS-PAGE Analysis of Histones and MBP Hydrolysis

Polyclonal unseparated IgGs from the blood sera of HIV-infected patients as shown earlier [27,28,29,30] and patients with MS effectively split both five human histones [31] and MBP [21,22]. Moreover, IgGs of HIV-infected patients against MBP and the five histones possess polyspecific complex formation and enzymatic cross-reactivity in the hydrolysis of five histones and MBP [32,33,34]. It was interesting to identify whether IgGs of MS patients against histones could also split both histones and MBP and vice versa. To analyze a possible enzymatic cross-reactivity, we first used the fraction of anti-histone IgGs (eluted from histone5-Sepharose) and anti-MBP IgGs (eluted from MBP-Sepharose). Figure 1A demonstrates the hydrolysis of all five histones by anti-histones and anti-MBP IgGs, while Figure 1B shows the hydrolysis of MBP by these IgGs.

Electrophoretically homogeneous human myelin basic protein preparations, unfortunately, were not available. Due to cDNA alternative splicing, as well as MBP forming products of partial hydrolysis in the different human brains, MBP preparations could consist of several forms (18.5, 17.5, and ≤14.0 kDa) and these protein hydrolysis products [21,22]. Figure 1B demonstrates myelin basic protein splitting by IgGs-abzymes against MBP and five histones. Line C in Figure 1B shows the essential heterogeneity of starting MBP preparation containing mostly the 18.5 kDa protein form. After 12 h of incubation with IgGs against histones and MBP, the relative content of all MBP forms decreased remarkably compared to the control experiment (lane C).

### 3.5. DNase Activity of IgGs against DNA and Proteins

The analysis was made to reveal the possibility of DNA hydrolysis with antibodies against DNA, five histones, and MBP. As an example, Figure 2 shows the data of the analysis of scDNA hydrolysis by several IgGs.

It can be seen that after 3 h of incubation with IgGs (10 µg/mL) against DNA, scDNA is completely hydrolyzed to the linear form of DNA (two or more breaks per molecule) and oligonucleotides of different lengths. IgGs against five histones and MBP effectively hydrolyzed scDNA during this time by about 25–35%, with the formation of a relaxed (one break per molecule) and linear form of DNA. An increase in the IgG concentration against MBP and five histones and the incubation time results in the formation of short splitting products, as in the case of IgGs against DNA. These data indicated that, as in the case of monoclonal antibodies against MBP obtained by Phage [54,55,56], the active centers of some monoclonal antibodies in the fractions of polyclonal MS patients’ IgGs against histones and MBP could combine two activities—protease and DNase.

These data of Figure 1 and Figure 2 may potentially demonstrate that anti-MBP, anti-histones and anti-DNA IgGs of MS patients could possess a known phenomenon of partially unspecific complex formation (polyreactivity or polyspecificity) revealed earlier [46,47,48,49], and they recognize all these molecules and hydrolyze them. These findings, however, cannot provide truthful evidence of enzymatic cross-reactivity between IgG-abzymes of MS patients against five histones, MBP, and DNA because, even after their isolations using several affinity chromatographies, it cannot be excluded that the obtained antibodies nevertheless could contain very small admixtures of alternative IgGs. The best proof of enzymatic cross-reactivity may be achieved from an undeniable difference in the specific sites of histone hydrolysis by IgGs against MBP, histones, and DNA. However, the principal objective of this study was not only to reveal the possibility of cross-hydrolysis of histones, MBP, and DNA by IgGs against these antigens but, first of all, to find out whether there is enzymatic cross-activity of Abs against other histones, MBP and DNA. This study first analyzed the possibility of the hydrolysis of histone H2A with specific IgG-abzymes against H1–H4 five histones, MBP, and DNA in order to understand how such multifunctional abzymes with several activities can be produced by B lymphocytes.

### 3.6. MALDI Analysis of H2A Histone Hydrolysis

The fractions of IgGs having high affinity to five individual histones (H1–H4), MBP, and DNA were used to find the cleavage sites of H2A by MALDI TOFF mass spectrometry. After the addition of the IgGs at zero time (Figure 3A), H2A histone was almost homogeneous, demonstrating only signals of its one- (*m*/*z* = 13981.9 Da) and two-charged ions (*m*/*z* = 6991.0 Da).

H2A cleavage assays were carried out with IgGs against this histone after 3–20 h of incubation. Many pronounced peaks corresponding to various sites of H2A histone splitting by IgGs against H2A have appeared after 3 h of hydrolysis (Figure 3B). Incubation of mixtures for 6–10 h led to the detection of several additional peaks (Figure 3C,D). After 20 h of incubation, H2A histone was completely hydrolyzed to very short peptides (Figure 3E). Based on the analysis of peaks in 9–10 spectra corresponding to different incubation times, twelve sites of H2A hydrolysis were identified, five of which are major, two moderate, and five minor splitting sites.

One of the exciting queries was whether IgGs of MS patients against H1, H2A, H3, and H4 histones could hydrolyze H2A histone. A very unexpected result was obtained after incubation of H2A with IgGs against histone H1. As shown in Figure 4A, even after 5 min of incubation of H2A with Abs against H1, there was significant hydrolysis of H2A histone.

After 3 h of incubation, H2A was completely hydrolyzed by antibodies against H1 (Figure 4B). The most significant number (19) of hydrolysis sites was found in this case.

Figure 4C–F show MALDI spectra of the hydrolysis products of H2A histone with anti-H2B IgGs for 3–20 h. Based on the analysis of 10 spectra for each incubation time, 12 splitting sites were detected.

Anti-H3 histone antibodies also hydrolyzed H2A (Figure 5A–D).

A total of 15 well detectable cleavage sites were found. Overall, 20 h of reaction mixture incubation resulted in the complete hydrolysis of histone H2A by anti-H3 histone IgGs to small peptides. (Figure 5D). Nine well-detectable sites were found for H2A hydrolysis with IgGs against H4 histone (Figure 5E).

All sites of H2A hydrolysis by different IgGs are shown in Figure 6.

As previously shown, antibodies against MBP from the blood of HIV-infected patients effectively hydrolyze all five histones, including H2A [32,33,34]. IgG antibodies against MBP of MS patients also hydrolyzed H2A histone but very slowly. The hydrolysis products were detected only after 20 h of the reaction mixture incubation. Figure 7 shows the spectrum of H2A hydrolysis by IgGs against MBP.

Abs against MBP hydrolyzed H2A histone at ten sites (Figure 6).

As a result of cell apoptosis, complexes of histones with DNA appear in the blood. These complexes can be very different, including partially dissociated ones. This can lead to the fact that antibodies against histones and DNA can be produced either against protein sequences of individual histones or against antigenic determinants forming at the junction of several different histones. In addition, anti-DNA antibodies can also be generated directly against DNA in its complexes with histones. However, at the junction of histones and DNA sequences, specific chimeric antigenic determinants can be formed, the production of antibodies against which can lead to the formation of variable regions of antibodies for the recognition of both proteins and DNA. Such pathways for antibody production can lead to the formation of chimeric polyfunctional molecules of Abs that recognize and hydrolyze both several histones and DNA. With this in mind, we analyzed the hydrolysis of histone H2A not only with IgGs against histones but also against DNA.

Figure 7B demonstrates MALDI spectra H2A hydrolysis with IgGs against DNA. The number of peaks corresponding to the hydrolysis of H2A, in this case, was remarkably greater than in the case of hydrolysis of this histone with IgGs against five histones and MBP. Based on the analysis of hydrolysis products, 23 sites of H2A hydrolysis by anti-DNA antibodies were found (Figure 6G). To facilitate comparing hydrolysis sites with different antibodies, they are shown in Table 1.

Figure 6 and Table 1 demonstrate that IgGs against five histones, MBP, and DNA remarkably differ in the cleavage efficiency, number, and positions of hydrolysis sites in the histone H2A protein sequence. Only antibodies against H2A, H2B, and MBP do not hydrolyze H2A histone in its C-terminal zone (L83-K129). A unique cluster of five H2A hydrolysis sites (L55-N68), four of which are major and one average, were found in the case of IgGs versus H1 histone (Table 1). Only in the case of H2A hydrolysis by antibodies against H2B were seven major cleavage sites found. Of the 10 sites of H2A splitting by antibodies against MBP, only one major and one average site was found, and the remaining eight were minor (Table 1).

Interestingly, the highest rate of H2A hydrolysis is observed for Abs against H1. At the same time, out of 19 sites of H2A hydrolysis by IgGs against H1, only 2 sites coincide with the sites of H2A splitting by antibodies against H2A (Table 1). For IgGs against H2A and other Abs, there are coincidences of different sites: 6 (anti-H2B and H3), 5 (anti-H4), and 4 (MBP) (Table 1). The location of different sites of H2A hydrolysis in the case of IgGs against H1, H2A, H2B, H3, H4, and MBP in different clusters of H2A histone and the different efficiency of their hydrolysis by antibodies against these five antigens indicate that they have not only polyreactivity of complex formation but also catalytic cross-reactivity.

Unlike antibodies against the four histones, no hydrolysis sites were found in the N-terminal zone of H2A (S1-H31) for IgGs against MBP. Interestingly, 5 hydrolysis sites in this zone were found for anti-DNA antibodies (Table 1). In the presence of significant differences, almost all IgGs hydrolyzed H2A at only four of the same sites: R32-L33, R35-K36, R77-I78, and R81-H82. A feature of the hydrolysis of H2A by IgGs against DNA is that they hydrolyze this histone at sites characteristic of Abs against different histones (Table 1). Moreover, the five sites of H2A hydrolysis by IgGs against DNA were not found in the case of Abs against five histones (Table 1).

## 4. Discussion

The polyreactivity of complex formation of various Abs is a widespread phenomenon [46,47,48,49]. Some compounds structurally similar to specific antigens can form complexes with Abs against a specific antigen. Therefore, during affinity chromatography, antibodies against specific antigens and Abs to some different compounds possessing structural elements of this antigen can form complexes with the immobilized cognate antigens [46,47,48,49]. The affinities of Abs for unspecific compounds are usually significantly lower than for specific antigens. Abs bound with unspecific molecules can generally be eluted during affinity chromatography using NaCl at a concentration comparable to 0.1–0.15 M [1,2,3,4,5,6,39,40,41]. Therefore, for obtaining IgGs against five individual histones (H1–H4), MBP and DNA, we eluted IgGs bound nonspecifically using 0.2 M NaCl. For additional purification of the IgGs against five histones and MBP, they were additionally passed using alternative affinity sorbents. Finally, IgG fractions were obtained against MBP, five individual H1–H4 histones, and DNA.

As shown in [31], IgG preparations of MS patients used by us in this study do not contain any classical proteinases or DNases. In addition, the same conclusion could be drawn based on the comparison of histone H2A splitting sites in the case of IgGs against five histones and MBP. Trypsin cleaves various proteins after the residues of lysine (K) and arginine (R). The H2A sequence contains 13 lysine and 13 arginine residues: 26 sites are potential cleavage sites of this histone by trypsin. However, the number of H2A hydrolysis sites by all 5 IgG preparations used after R and K varies mainly but is significantly lower than 26. In addition, hydrolysis after the K and R residues occurs only in specific clusters of H2A (Figure 6, Table 1).

Chymotrypsin hydrolyzes various proteins after aromatic amino acids (F, Y, and W). There are four potential such sites for hydrolysis in H2A histone by chymotrypsin. Only one site of hydrolysis after Y was found for anti-H1 IgGs (Y57-L58), but not a single hydrolysis site was found after F (Table 1). The cleavage sites of H2A by different IgGs occur basically in clusters after neutral non-charged and nonaromatic amino acids: S, G, E, L, A, P, Q, and T (Figure 6, Table 1). Thus, the sites of specific hydrolysis of H2A by various IgGs do not correspond to trypsin or chymotrypsin and are located in specific amino acid clusters, not distributed along the entire length of the H2A histone molecules.

Using many monoclonal Abs of SLE patients, it was shown that their active centers could correspond to serine, thiol, or metal-dependent proteases [50,51,52,53,54,55,56]. In contrast to classical proteases, abzymes split specific proteins mainly in their specific amino acid clusters [1,2,3,4,5,6,27,28,29,30,31,32,33,34].

One intriguing result is that IgGs of MS patients not only against histone H2A but also against H2B, H3, H4, MBP, and DNA are able to hydrolyze histone H2A. The primary evidence that the preparations of each IgG to MBP and five individual histones do not contain at least relevant for analysis impurities of IgGs against any of the histones or MBP is that H2A cleavage sites for each of them are significantly different (Figure 6, Table 1). Using IgGs against each of five histones and MBP from the blood of HIV-infected patients, it was shown that the main reason for enzymatic cross-reactivity might be a high level of homology of the protein sequences of MBP and five histones [32,33,34]. Therefore, it was interesting to analyze the level of homology between the protein sequence of H2A with three other histones and MBP.

The complete identity of amino acids between H2A and H1 and three alignments was from 26.2 to 27.1% (average value 26.7 ± 0.5%), while similarity (identical together with non-identical amino acids but with highly similar physicochemical properties) varied from 49.4 to 54.4% (average 52.7 ± 2.9%). Identity between H2A and H2B according to three different alignments changes from 26.7 to 30.5% (average value 28.5 ± 3.0%) and similarity from 47.0 to 60.3% (average 51.6 ± 7.6%).

For H2A and H3 histones, the following homology values were found: identity 28.3–28.7% (average value 28.6 ± 0.5%); similarity 52.6–54.8% (average value 53.4 ± 1.2%).

H4 has 32.8% identity and 53.8% similarity with H2A.

An analysis of the homology between the complete sequences of MBP and five histones was carried out: H2A (identity—25.0–26.8% (average 25.9 ± 1.3%), similarity 47.6–50.3% (average 49.0 ± 1.9%)); H1 (identity—25.4–28.4% (average 26.9 ± 2.1%), similarity 48.8–52.8% (average 50.8 ± 2.8%)); H2B (identity—24.4–25.4% (average 25.4 ± 2.2%); similarity—46.0–53.2% (average value 49.1 ± 3.5%)), H3 (identity—22.8–25.3% (average24.4 ± 1.4%), similarity 44.3–47.6% (average 45.5 ± 1.8%)); H4 (identity—21.3–30.0% (average 25.5 ± 4.6%); similarity—45.0–55.6% (average value 55.1 ± 5.5%)).

However, more likely than not, the general homology between the complete sequences of the proteins, but the homology between the sequences hydrolyzing by various abzymes, may be more important for the manifestation of abzyme enzymatic cross-reactivity.

It should be emphasized that all histones and MBP contain many positively charged residues of lysine and arginine. These amino acid residues are necessary to histones for their interaction with negatively charged internucleoside groups of DNAs. Moreover, it was demonstrated that MBP could efficiently form complexes with DNAs [59]. Thus, it is possible that many positively charged amino acids in MBP and five histones could also make an essential contribution to the ability of IgGs against these proteins to form complexes with foreign histones and MBP and hydrolyze these proteins. At the same time, it cannot be excluded that some abzyme-autoantibodies are formed against chimeric antigenic determinants consisting of protein sequences, for example, of two or more histones. This may also be the reason for the hydrolysis of H2A (and possibly other histones) by IgGs having an increased affinity for any of the five histones and MBP.

A particularly interesting situation is observed for IgGs against DNA. These Abs show a particularly high number of H2A hydrolysis sites (Table 1). It is important that different sites of H2A hydrolysis by anti-DNA IgGs were found for Abs against five different histones and MBP. However, DNA can be in complexes with various individual histones and their associates. As a result, the pool of polyclonal IgGs with a high affinity for DNA-cellulose can consist of antibodies to a wide variety of chimeric antigenic determinants, consisting of DNA and any of the five individual histones or even their associates. This way of producing abzymes can provide an exceptional variety of antibodies, in the active centers of which are combined the active centers similar to DNases and proteases, which can hydrolyze DNA and different histones.

As mentioned above, abzymes against histones, MBP and DNA are toxic. However, the combination of several catalytic activities in some antibody molecules can lead to a significant increase in their toxicity. Since anti-DNA antibodies easily penetrate through the cell and nuclear membranes, chimeric antibodies can hydrolyze not only DNA but also chromatin histones.

In addition, histone complexes with DNA that have entered the blood as a result of cell apoptosis can significantly stimulate the production of abzymes. Anti-histone and chimeric anti-histone-DNA antibodies can hydrolyze MBP in sheath neural tissues, promoting the development of multiple sclerosis. Thus, such abzymes may play an important role in MS pathogenesis.

## 5. Conclusions

Here, we have first shown using abzymes from patients with MS that IgGs against H2A, H1, H2B, H3, H4, MBP, and DNA possess an ability similar to anti-H2A IgGs to form complexes with H2A histone, demonstrating polyreactivity in complexation. Moreover, an exciting result was obtained. IgG-abzymes against H2B, H1, H3, H4, myelin basic protein, and DNA possess catalytic cross-reactivity with anti-H2A antibodies, and all of them are capable of hydrolyzing histone H2A. Evidence that the ability of IgGs against H2A, H2b, H1, H3, H4, MBP, and DNA to hydrolyze H2A histone is their property follows from the fact that the sites of hydrolysis of H2A histone are different for IgGs against H2A, H2b, H1, H3, H4, MBP, and DNA, including their location in the H2A protein molecule. The formation of abzymes hydrolyzing H2A in the IgG pool is most probably associated with a high sequence homology of all histones and MBP. The production of abzymes with high affinity for DNA, hydrolyzing not only DNA but also all five histones and MBP, is most likely a consequence of the formation of chimeric antigenic determinants containing DNA-protein complexes proteins and DNA sequences. Since all histones constantly occur in human blood due to cell apoptosis, the existence of catalytic cross-reactivity of IgGs- abzymes against histones, MBP and DNA can play a very negative role in MS pathogenesis.

## Figures and Tables

**Figure 1 biomedicines-10-01876-f001:**
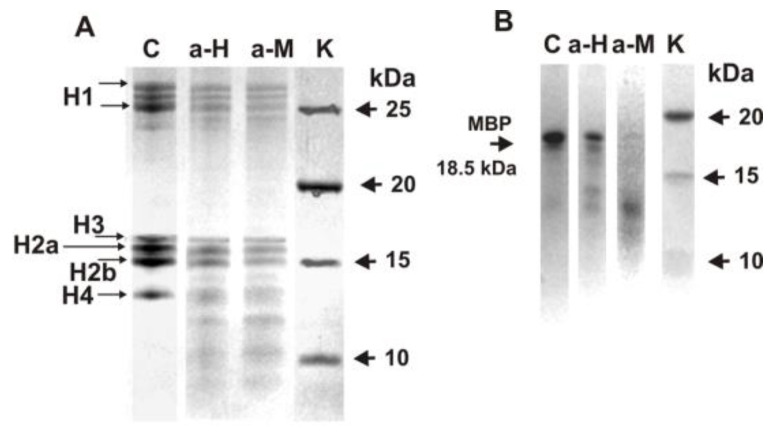
SDS-PAGE analysis hydrolysis of five histones (H1–H4) by IgGs-abzymes against these five histones (lane a-H) as well as MBP (lane a-M) (**A**) and human myelin basic protein by IgGs against five histones (lane a-H) and Abs-abzymes against MBP (lane a-M) (**B**). MBP and a mixture of five histones with and without IgGs (0.03 mg/mL) were incubated for 12 h. Lane C corresponds to the histones (**A**) and MBP (**B**) incubated without IgGs. Lane K—proteins with known molecular masses (**A**,**B**).

**Figure 2 biomedicines-10-01876-f002:**
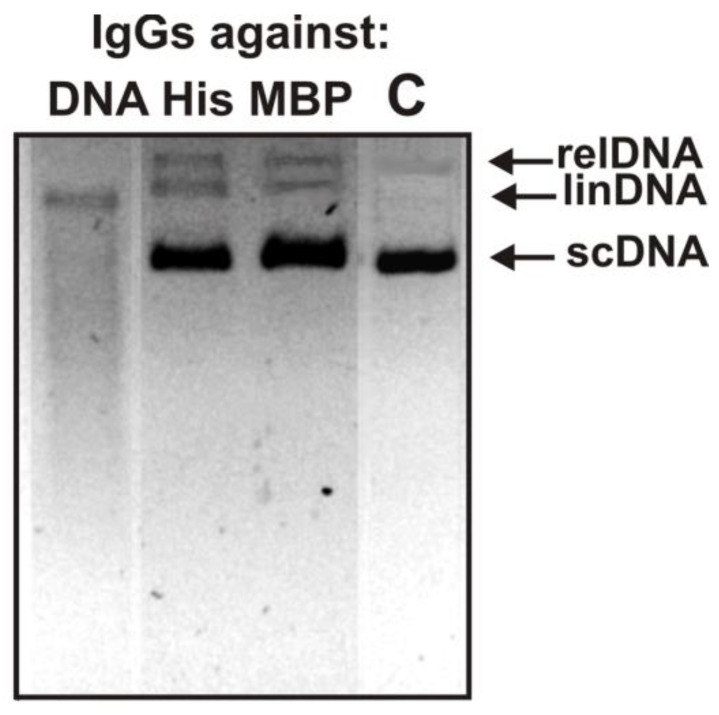
Analysis of the relative activity of IgGs against DNA, five different histones, and MBP in the hydrolysis of scDNA for 3 h. Designations of IgGs are shown in the Figure.

**Figure 3 biomedicines-10-01876-f003:**
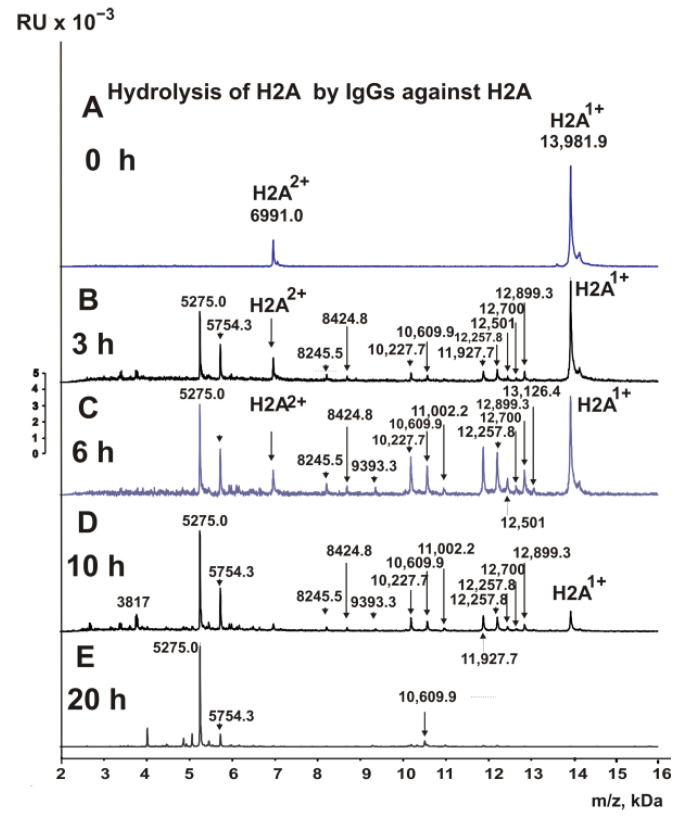
MALDI spectra show H2A histone (0.8 mg/mL) over time during hydrolysis (0–20 h) in the presence of IgGs (0.045 mg/mL) against H2A (**A**–**E**).

**Figure 4 biomedicines-10-01876-f004:**
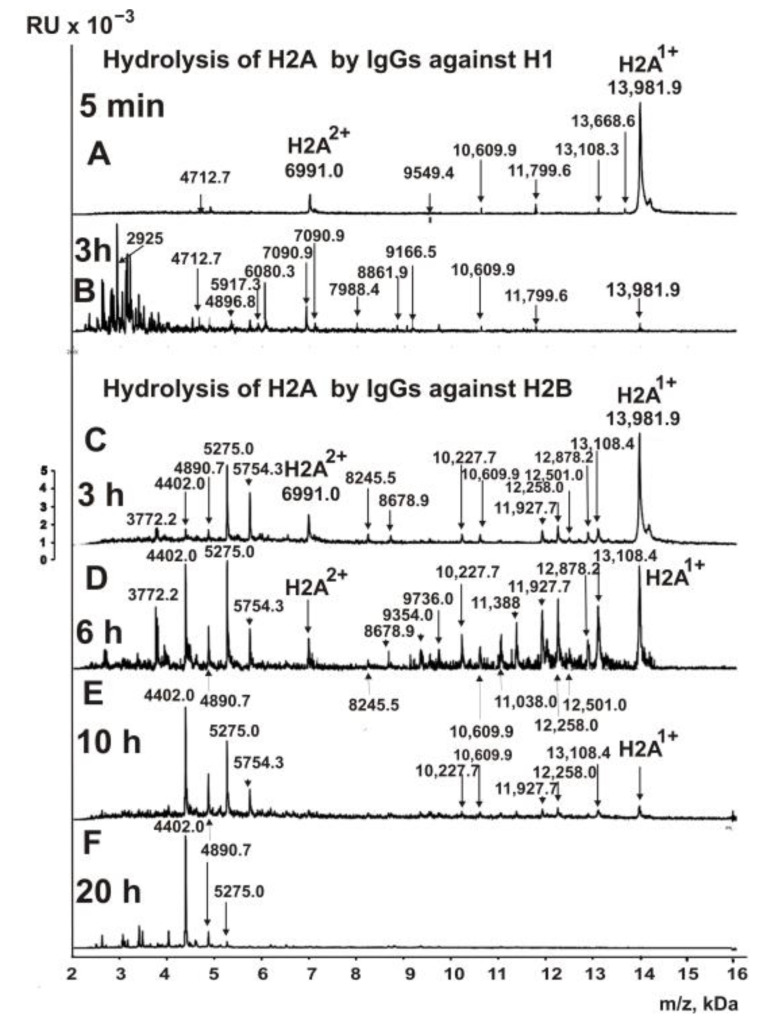
MALDI spectra corresponding to H2A histone (0.8 mg/mL) over time during hydrolysis in the presence of IgGs (0.04 mg/mL) against H1 (**A**,**B**) and H2B (**C**–**F**).

**Figure 5 biomedicines-10-01876-f005:**
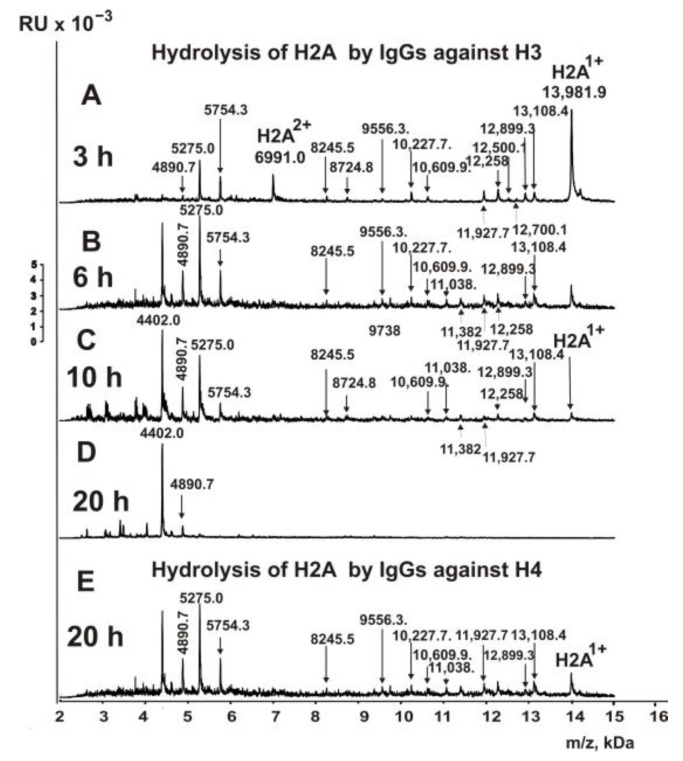
MALDI spectra demonstrating in time hydrolysis of H2A by IgGs (0.04 mg/mL) against H3 histone (**A**–**D**) and H4 histones (**E**).

**Figure 6 biomedicines-10-01876-f006:**
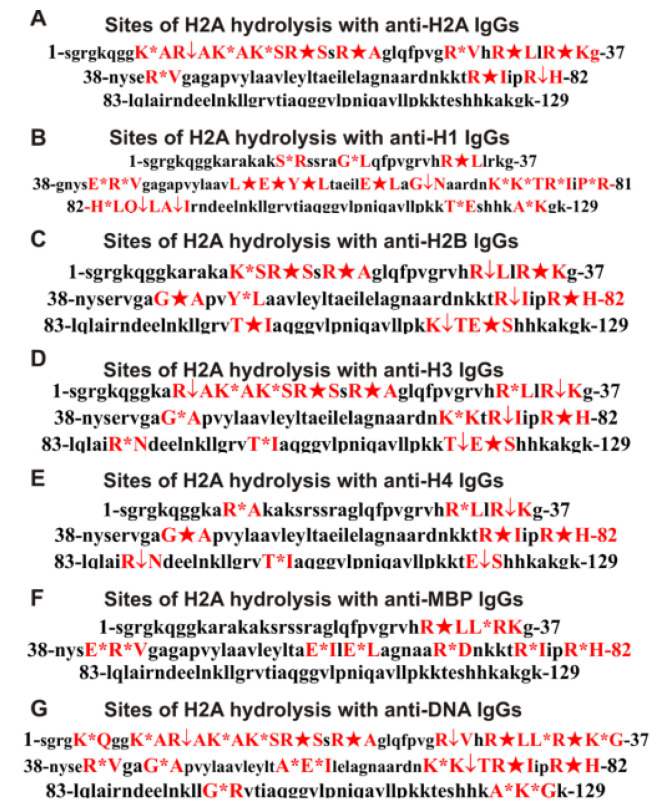
Sites of H2A hydrolysis by IgGs against H2A (**A**), H1 (**B**), H2B (**C**), H3 (**D**), H4 (**E**), MBP (**F**), and DNA (**G**). Major sites of H2A splitting are marked by big stars (★), moderate ones by arrows (↓), and minor sites of the cleavages by small stars (*) (**A**–**F**).

**Figure 7 biomedicines-10-01876-f007:**
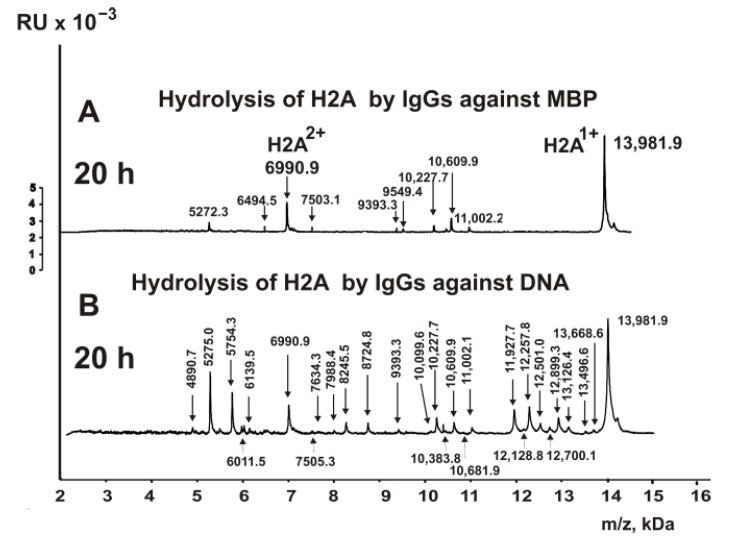
MALDI spectra demonstrate in time hydrolysis of H2A by IgGs (0.04 mg/mL) against MBP (**A**) and DNA (**B**).

**Table 1 biomedicines-10-01876-t001:** Sites of H2A histone hydrolysis by Abs against five histones, MBP, and DNA.

Type of IgGs						
Anti-H2A	Anti-H1	Anti-H2B	Anti-H3	Anti-H4	Anti-MBP	Anti-DNA
12 Sites *	19 Sites	11 Sites	12 Sites	9 Sites	10 Sites	23 Sites
-	-	-	-	-	-	K5-Q6
K9-A10	-	-	*-*	-	-	K9-10A
* **R11-A12** *	-	-	* R11-A12 *	R11-A12	-	* R11-12A *
K13-14A	*-*	-	K13-A14	-	-	K13-14A
K15-16S	-	K15-S16	K15-S16	-	-	**K15-S16**
-	S16-R17	-	-	-	-	-
**S16-R17**	-	-	-	-	-	-
-	-	R17-S18	R17-S18	-	-	R17-S18
-	G22-L23	-	-	-	-	-
**R20-A21**	-	**R20-A21**	**R20-A21**	-	-	**R20-A21**
**R** **29** **-30V**	-	-	-	-	-	** *R* ** ** *29* ** ** *-30V* **
**R32-L33**	**R32-L33**	* R32-L33 *	R32-L33	** *R* ** ** *32* ** ** *-L33* **	**R32-L33**	**R32-L33**
**-**	-	* - *	-	*-*	-	**L34-R33**
**-**	-	*-*	-		L34-R35	-
**R35-K36**	-	**R35-K36**	* **R35-K36** *	**R35-K36**	** *R35-R36* **	**R35-K36**
**-**	-	-	*-*	**-**	*-*	**K36-37G**
**-**	E41-R42	-	-	**-**	E41-R42	-
**R** **42** **-43V**	-	-	-	**-**	R42-V43	**R** **42** **-43V**
-	V43-G44	-	-	**-**	-	-
-	-	**G46-A47**	G46-A47	**G46-A47**	-	G46-A47
-	-	Y50-L51	-	-	-	-
-	**L55-E56**	-	-	-	-	-
-	**E56-Y57**	-	-	-	-	-
-	**Y57-L58**	-	-	-	-	-
	**-**	-	-	-	-	**A60-E61**
-	**-**	-	-	-	E61-I62	E61-I62
-	**E64-L65**	-	-	-	E64-L65	-
-	** *G67-N68* **	-	-	-	-	-
-	*-*	-	-	-	R71-D72	
-	K74-K75	-	K74-K75	-	-	K74-K75
-	*K75-T76*	-	-	-	-	*K75-T76*
**R77-I78**	R77-I78	*R77-I78*	**R77-I78**	**R77-I78**	R77-I78	**R77-I78**
**-**	-	*-*	**-**	-	-	-
**-**	P80-R81	-	**-**	-	-	-
** *R81-H82* **	-	**R81-H82**	**R81-H82**	**R81-H82**	R81-Y82	**R81-H82**
*-*	H82-L83	**-**	-	-	-	-
-	** *Q84-L85* **	** *-* **	-	-	-	-
-	** *A86-I87* **	**-**	-		-	-
-	*-*	**-**	R88-N89	R88-N89	-	-
		**-**	-	-	-	G98-R99
-	*-*	**T101-I102**	T101-I102	T101-I102	-	-
-	*-*	** *K119-T120* **	-	-	-	-
-	T120-E121	**-**	** *T120-E121* **	-	-	-
-	-	**E121-S122**	**E121-S122**	* **E121-S122** *	-	-
-	A126-K127	-	-	-	-	A126-K127
-	-	-	-	-	-	K127-G128

* Major hydrolysis sites are marked in bold (red), moderate in italics (green), and minor sites in regular (black): missing hydrolysis sites are marked with a dash (-).

## Data Availability

The data supporting our study results are included in the article and Appendix A.

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
