# Peer review of "Multiple Sclerosis: Enzymatic Cross Site-Specific Recognition and Hydrolysis of H2A Histone by IgGs against H2A, H1, H2B, H3 Histones, Myelin Basic Protein, and DNA"

_biomedicines, 2022, doi:10.3390/biomedicines10081876_

Round 1

Reviewer 1 Report

This MS is informative, and well written. 

Minor comments: 

Add the hypothesis and define the objectives in the introduction section 

Could you please clarify the legend for the figures 1 and 2. 

could you point out the limits of the study

Could you deeply discuss why this study is important in this context 

Thank you 

Author Response

This MS is informative, and well written. 

Minor comments: 

Add the hypothesis and define the objectives in the introduction section 

Could you please clarify the legend for the figures 1 and 2. 

could you point out the limits of the study

Could you deeply discuss why this study is important in this context 

Answer:

We added information in legends to Figures

Figure 1. SDS-PAGE analysis hydrolysis of five histones (mixture of five histones H1, H2A, H2B, H3 and H4 was used for analysis) by IgGs-abzymes against these five histones (lane a-H) as well as MBP (lane a-M) (A) and splitting of human myelin basic protein by IgGs against five histones (lane a-H) and Abs-abzymes against MBP (lane a-M) (B). MBP and a mixture of five histones with and without IgGs (0.03 mg/ml) were incubated for 12 h. Lanes C correspond to the histones (A) and MBP (B) incubated without IgGs. Lanes K - proteins with known molecular masses (A and B).

Figure 2. Analysis of the relative activity of IgGs against DNA, five different histones, and MBP in the hydrolysis of scDNA for 3 h. Designations of IgGs are shown in the Figure. IgGs against MBP, five histones and DNA were obtained by affinity chromatographies of MBP-Sepharose, histone5-Seharose and DNA-cellulose and used for analysis of scDNA hydrolysis. Different forms of DNA are shown: non-hydrolyzed supercoiled DNA (scDNA), relaxed DNA (relDNA; one break in the DNA), and linear form of DNA (linDNA; several breaks in DNA).    

In the end of discussion we have added:

As noted above, it is believed that the development of AI reactions can be stimulated by different viral or bacterial infections and production of Abs against viral proteins, and then after the immune system is disrupted, it begins to synthesize auto-antibodies against human proteins. [1-6].  Since antibodies against complexes histones with DNA are capable of hydrolyze MBP and vice versa, the catalytic polyreactivity of antibodies against these antigens provides internal factors for the development of multiple sclerosis.

Many thanks for the kind comments

With best regards

Prof. Georgy Nevinsky

Reviewer 2 Report

Please refer your manuscript to an extensive editing of English language and style, before to resend it.

Author Response

Please refer your manuscript to an extensive editing of English language and style, before to resend it.

Answer. We tried to check English

Many thanks for the kind comments

With best regards

Prof. Georgy Nevinsky

Round 2

Reviewer 2 Report

The manuscript entitled "Multiple sclerosis: enzymatic cross site-specific recognition 1 and hydrolysis of H2A histone by IgGs against H2A, H1, 2 H2B, H3 histones, myelin basic protein, and DNA" has to be fundamentally improved.

First of all the patient's group is to heterogeneously according with disease duration to have any valuable conclusions regarding the associated inflammatory process. I suggest to select more patients and to group them according to disease duration. Demyelination and remyelination processes are associated with inflammatory and inflammation remission episodes. Therefore, since the disease duration is important for the number of these episodes, at least as the clinical form of MS is, please select more patients for your study. Moreover, indicate the correct classification of the MS: clinical isolated syndrome, relapsing remitting forms etc , and the a sufficient number of the patients corresponding to each clinical form, to have some correct statistical results and conclusions.

Extensive English editing is still needed. 

Author Response

The manuscript entitled "Multiple sclerosis: enzymatic cross site-specific recognition 1 and hydrolysis of H2A histone by IgGs against H2A, H1, 2 H2B, H3 histones, myelin basic protein, and DNA" has to be fundamentally improved.

First of all the patient's group is to heterogeneously according with disease duration to have any valuable conclusions regarding the associated inflammatory process. I suggest to select more patients and to group them according to disease duration. Demyelination and remyelination processes are associated with inflammatory and inflammation remission episodes. Therefore, since the disease duration is important for the number of these episodes, at least as the clinical form of MS is, please select more patients for your study. Moreover, indicate the correct classification of the MS: clinical isolated syndrome, relapsing remitting forms etc , and the a sufficient number of the patients corresponding to each clinical form, to have some correct statistical results and conclusions.

Extensive English editing is still needed. 

Sorry, but this article is aimed at finding out what kind of antibodies with cross-recognition and cross-catalysis can in principle be formed in MS patients. In this first study of the fundamental disclosure of possible types of new unusual antibodies was performed. Here, we did not try to compare a possible ratio of such antibodies against MBP and five histones, hydrolyzing H2A histone and anti-DNA hydrolyzing MBP and H2A histone in different types of MS patients and  can at different stages of multiple sclerosis development of. Thus, we have shown that such antibodies can, in principle, can be produced in patients with MS.

What you propose to do next is a new and very laborious work. Such new work can be carried out in the future, already taking into account the data obtained in this study. These data give an idea of ​​what kind of activity to look for in the course of disease development at different stages of MS. We plan to carry out such work, but its implementation will require a long time of accumulation of antibody preparations corresponding to different stages of MS and a long analysis of the abnormal catalytic antibodies found in this work. However, from our point of view, you propose to do another new work, you propose to do another new work that cannot be done properly without the data of this first article.

Thank you for your comment, we plan to conduct the study you suggested in the future.

Sincerely

Prof. Geogy A. Nevinsky